# LIFE: A Flexible and Efficient Module for Incorporating Local Information into Vision Transformers

## Abstract

Vision transformers (ViTs) achieve remarkable performance on large datasets, but tend to perform worse than convolutional neural networks (CNNs) when trained from scratch on smaller datasets, possibly due to a lack of local inductive bias in the architecture. Recent studies have therefore added locality to the architecture and demonstrated that it can help ViTs achieve performance comparable to CNNs in the small-size dataset regime. Existing methods, however, are architecture-specific or have higher computational and memory costs. Thus, we propose a module called *Local InFormation Enhancer (LIFE)* that extracts patch-level local information and incorporates it into the embeddings used in the self-attention block of ViTs. Our proposed module is memory and computation efficient, as well as flexible enough to process auxiliary tokens such as the classification and distillation tokens. Empirical results show that the addition of the LIFE module improves the performance of ViTs on small image classification datasets while preserving performance on large-scale datasets. We further demonstrate how the effect can be extended to downstream tasks, such as object detection and semantic segmentation. In addition, we introduce a new visualization method, Dense Attention Roll-Out, specifically designed for dense prediction tasks, allowing the generation of class-specific attention maps utilizing the attention maps of all tokens.[1]

## 1 Introduction

Transformers, a new kind of encoder-decoder model that uses a self-attention mechanism to process input data (Vaswani et al., 2017), were initially proposed for sequence modeling in the natural language processing (NLP) domain. The success of transformers in NLP has led to the development of these architectures for a wide range of vision tasks (Dosovitskiy et al., 2020; Liu et al., 2021b; Jeeveswaran et al., 2022). Vision transformers (ViTs), when trained or pre-trained on a large dataset, outperform their CNN counterparts. However, for many real-world vision tasks, a large amount of annotated data is either too expensive or not feasible. As a result, the data-hungry nature of ViTs prevents them from being applied to a number of crucial real-world problems for which a limited amount of annotated data is available.

In majority of ViTs, an image is divided into a sequence of non-overlapping patches, from which the self-attention layer learns the global context. The information in patches that are spatially adjacent to a given patch can be used to create its local context. ViTs do not, however, exploit this information due to a low inductive bias that is only coming from strided convolution in patch embedding layer (Dosovitskiy et al., 2020). Convolutional layers, on the other hand, enable CNN architectures to utilize local information at the pixel level, thereby enhancing their data efficiency (He et al., 2016; Tan & Le, 2019). The local context may therefore be essential in enabling ViTs to learn vision tasks with fewer data samples.

Recent studies have improved the use of the local context in ViTs through architectural modifications (Liu et al., 2021b; dAscoli et al., 2021), token pre-processing (Yuan et al., 2021b), or the addition of convolutional layers (Chen et al., 2021). Despite the fact that these findings support the importance of utilizing local information, their design is not adaptable enough to be annexed to other ViT architectures and/or has a negative impact on other performance factors such as memory consumption and computational cost.

---

[1]The code will be publicly available upon acceptance.

Therefore, it is advantageous to devise a method to incorporate local information effectively and modularly into the architecture of ViTs.

The functionality of a transformer architecture depends on its self-attention layers, which require a sequence of embeddings. These embeddings are typically generated using a point-wise feedforward layer with a receptive field consisting of only one patch from the input image or one token from the previous layer (Figure 1(a)). However, the use of larger receptive fields can result in embeddings that contain local information from spatially adjacent patches, as shown in Figure 1(b). We hypothesize that by enriching the embeddings with this local context, vision transformer models (ViTs) will be able to learn the global context with fewer data points and perform better on smaller datasets.

We propose the LIFE (*Local InFormation Enhancer*) module to improve the performance of vision transformers on smaller datasets by creating embeddings for self-attention layers with larger receptive fields. The LIFE module reshapes the input tokens from the previous layer into an image format and applies convolution layers with multiple kernel sizes. After the feature maps from the convolutional layers are transformed back into tokens, they are sent to the self-attention layers. To add local context to all patch tokens and any other auxiliary tokens in the ViT architecture, we use depthwise separable convolutional (DSC) layers (Chollet, 2017) in the LIFE module. Unlike a standard convolution layer, a DSC layer performs the computations in two steps: a depth-wise convolution layer followed by a point-wise convolution layer, which is more computationally efficient. Note that auxiliary tokens, such as classification and distillation tokens, in ViTs are processed by the pointwise convolution layer in the DSC.

We evaluate the efficacy and versatility of the LIFE module by integrating it into different ViT architectures with varying capacities. Using DeiT, T2T, and Swin transformers as base architectures, we evaluate classification performance on the ImageNet-100, CIFAR10, CIFAR100, Tiny-ImageNet, and ImageNet-1k datasets. We also evaluate the addition of LIFE to ViTs on dense prediction tasks using the VOC dataset for object detection and the Cityscapes dataset for semantic segmentation tasks. Our extensive empirical experiments demonstrate that the LIFE module can be easily integrated into various ViT architectures and consistently improves performance, regardless of the task at hand, with negligible memory and computation overhead. In addition, we qualitatively assess the contribution of the LIFE module to each task. To visualize the attention for dense prediction tasks, we propose a dense attention roll-out. Our results further support the notion that LIFE can enhance local context learning by guiding the network to attend to more specific regions. The contributions of our work can thus be summarized as follows;

- Introducing *Local InFormation Enhancer (LIFE)* module, which complements the global information by adding local context to the embeddings used in ViT.
- Demonstration of the ability of the LIFE module to be easily integrated into different ViT architectures with minimal memory and computation costs overhead, even in the architecture that contains auxiliary tokens.
- Employing the LIFE module in different ViTs results in performance gains on smaller datasets such as ImageNet-100, Tiny-ImageNet, CIFAR10, and CIFAR100.
- LIFE module is versatile and our experiments demonstrate that the LIFE module can boost the performance of dense prediction tasks.
- Proposing a novel method, *Dense Attention Roll-Out*, to visualize attention for dense prediction tasks.
- Qualitative evaluation of the contribution of the LIFE module to each task using our proposed visualization method.

## 2 Related Work

**Vision Transformers (ViT)** have demonstrated competitiveness with convolutional neural networks (CNNs) in various vision tasks, such as image classification (Dosovitskiy et al., 2020; Touvron et al., 2021; Liu et al., 2021b), object detection (Carion et al., 2020), and semantic segmentation (Zheng et al., 2021;

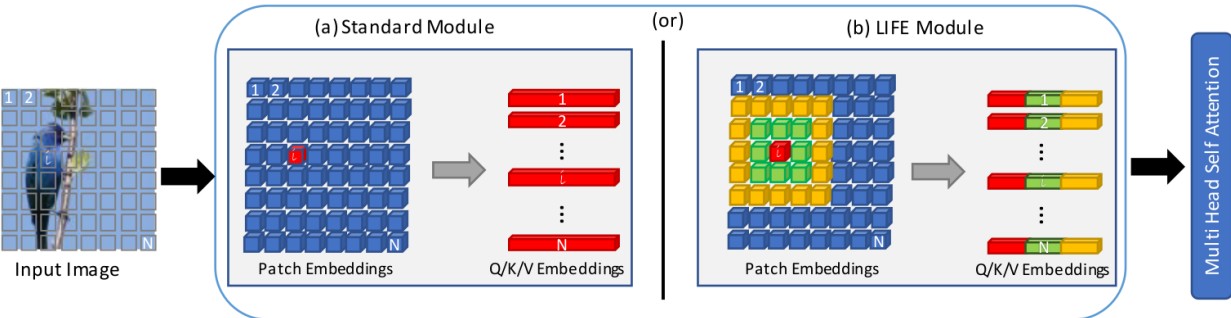

Figure 1: In vision transformers, the input image is divided into $N$ non-overlapping patches, which are then transformed into embeddings. (a) These patch embeddings are then passed through a pointwise feedforward layer to generate query (Q), key (K), and value (V) embeddings. These Q, K, and V embeddings are then input to a self-attention layer. (b) Alternatively, the LIFE module integrates local information from adjacent patches into the Q, K, and V embeddings by using sequentially larger receptive fields. For example, the standard module uses only the $i^{\text{th}}$ patch to generate the $i^{\text{th}}$ Q/K/V embedding, while the LIFE module includes information from adjacent patches (shown in green and orange) as well.

Cheng et al., 2021). These models utilize self-attention at the early levels to construct a convolution-free neural network. Following the introduction of the original ViT model (Dosovitskiy et al., 2020), numerous studies have focused on improving classification performance through architectural modifications, knowledge distillation, or advanced data augmentation techniques (Touvron et al., 2021; Han et al., 2021; Yuan et al., 2021b; Wang et al., 2021a; 2022; Liu et al., 2021b; 2022).

**Locality.** Convolutional neural networks (CNNs) have become a common architecture for visual tasks. They typically contain a stack of convolution layers with small kernel sizes that utilize information from neighboring feature vectors. This architectural design exploits the spatial correlation in the natural images, making CNNs more data efficient. Evidence from CNNs suggests that it is essential to utilize local information to improve performance on small-scale datasets (He et al., 2016; Tan & Le, 2019). In contrast, the self-attention mechanism in the transformer block establishes a global relation between tokens, but ignores the locality. To address this, many approaches have been proposed to introduce locality bias into transformer architectures through architectural modifications.

Several recent studies have proposed hybrid networks to incorporate locality bias from convolution operations into transformer architectures. CeiT (Yuan et al., 2021a) uses low-level features from CNNs rather than patches extracted from raw images, and introduces a depth-wise separable convolution into the feedforward network within the transformer block to improve locality. CvT (Wu et al., 2021) integrates convolution into token embeddings, allowing a progressively decreasing number of tokens while increasing the dimension of the features. It also uses a depth-wise separable convolution to compute query, key, and value. Crossformer (Wang et al., 2021b) addresses the lack of multiscale information in transformer architectures by processing tokens using short- and long-distance attention and combining multiscale information from neighboring tokens using multiple convolution layers with different kernel sizes within a pyramid-like architecture.

Another line of work proposes to incorporate a pyramid structure derived from CNNs into the transformer architecture in order to induce locality in the network. This is achieved through various techniques, such as the use of a gradual shrinking technique in PVT (Wang et al., 2021a; 2022), self-attention within windows in Swin (Liu et al., 2021b; 2022), and hierarchical aggregation of transformer blocks in NesT (Zhang et al., 2022). All of these approaches involve the gradual combination of neighboring tokens, enabling the network to consider the local context in its processing.

Other approaches aim to introduce locality while maintaining the pure transformer architecture. T2T (Yuan et al., 2021b) replaces the standard patch embedding layer with progressive tokenization to combine neighboring tokens; while TNT (Han et al., 2021) divides the input image into large patches called visual sentences

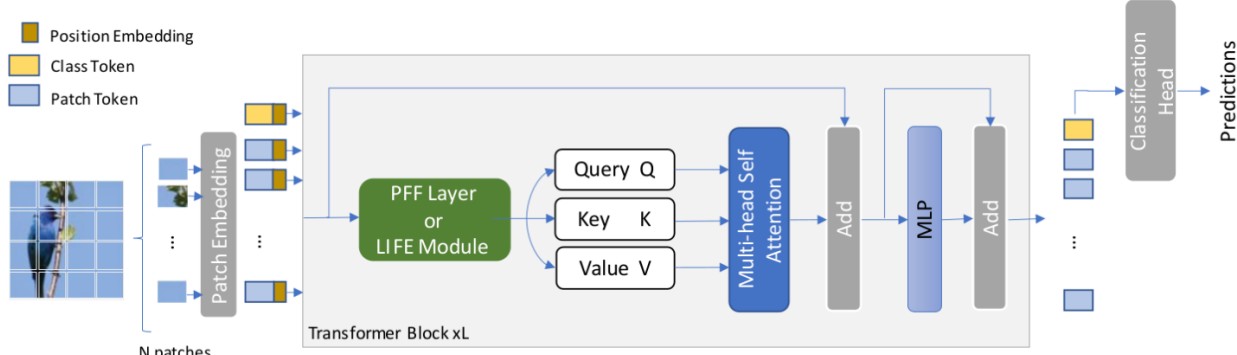

Figure 2: Architecture of a standard ViT. In order to incorporate local context in the embeddings for multi-head self-attention layers, we replace the pointwise feedforward (PFF) layer with our proposed LIFE module (details in Figure 3), where it generate multiscale query, key, and value.

and small patches called visual words. TNT uses a shared sub-transformer architecture to extract the relation and similarity between the small patches, and passes the information generated by the sub-transformer to the visual sentence for further processing by the standard transformer block.

However, the aforementioned approaches are architecture specific and focus on designing effective models for large-scale datasets, but may not necessarily perform well on smaller datasets. Liu et al. (2021a) propose a method that introduces a self-supervised task to predict the geometric distance between pairs of output tokens, while Li et al. (2022) propose a distillation-based training mechanism that uses a lightweight pre-training CNN model to distill its features into a ViT. Although these approaches are flexible and can be integrated into different architectures, they may also come with additional memory and computational costs, and may not be as effective when applied to smaller datasets.

On the contrary, the LIFE module is a modular and efficient approach that utilizes locality to generate richer embeddings, which is more effective for learning with fewer data. It introduces locality in query, key, and value, making it flexible for integration with any transformer architecture. Unlike CvT, the LIFE module benefits from multiscale information and is more efficient compared to T2T and Crossformer, as it uses dense concatenation and depthwise separable convolution. Additionally, unlike previous work, the LIFE module utilizes multi-scale locality in every block and can be used in conjunction with training mechanisms for small-scale datasets.

## 3 Methodology

In ViTs, an input image is divided into $N$ non-overlapping square patches (Touvron et al., 2021). These patches are then flattened and embedded in patch tokens $X_{patch}$ of length $C$ using a linear layer. Positional embeddings are added to the token patches and an additional classification token $X_{cls}$, which is a learnable embedding of the same length, forming a matrix $X$:

$$X = [X_{patch}|X_{cls}]; \qquad X \in \mathbb{R}^{C \times (N+1)}, X_{patch} \in \mathbb{R}^{C \times N}, X_{cls} \in \mathbb{R}^{C \times 1} \tag{1}$$

The resulting matrix is then passed to a series of $L$ transformer blocks (Figure 2), each consisting of a point-wise feedforward layer (PFF), followed by a multi-head self-attention layer (MHSA) and a multi-layer perceptron (MLP):

$$\begin{aligned} Q, K, V &\leftarrow PFF(X), \\ X_{out} &= MLP(MHSA(Q, K, V)), \end{aligned} \tag{2}$$

The final prediction is usually obtained by processing the classification token at the end of the last transformer block or by using global average pooling of the patch tokens. However, standard ViTs lack local inductive bias; therefore, to introduce local context, we replace the PFF in Eq. 2 with our proposed LIFE module.

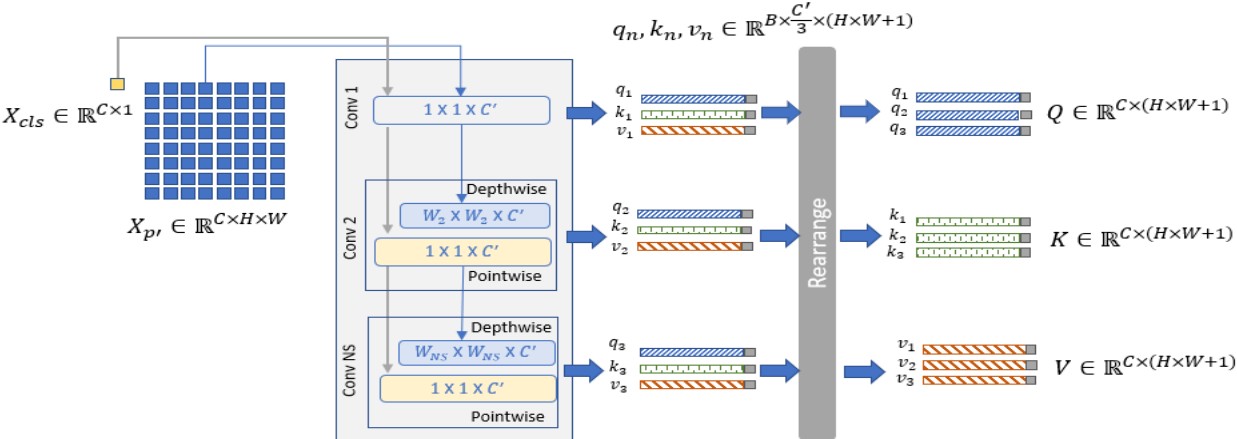

Figure 3: Overview of *LIFE* module. It consists of a hierarchy of convolutional layers, with the first layer being a pointwise convolution and the remaining layers being depthwise separable convolutions. Both the classification token $X_{cls}$ and patch tokens $X_{p'}$ (arranged in the input image format) are passed through these layers (marked in $\downarrow$ and $\downarrow$, respectively). Each layer in the hierarchy outputs local information from a relatively larger receptive field. The information is then divided into query $q_i$, key $k_i$, and value $v_i$ (including the processed class token represented as a gray box). These hierarchical features are then rearranged to form the final query $Q$, key $K$, and value $V$.

### 3.1 LIFE Module

To incorporate local information, the LIFE module uses multiple hierarchically arranged convolutional layers, as depicted in Figure 3. The first layer in the hierarchy is always a point-wise convolution with the smallest receptive field, while subsequent layers have progressively larger receptive fields due to increasing kernel sizes. The output features of these layers represent local information gleaned from various receptive fields, and are rearranged to obtain the final query $Q$, key $K$, and value $V$. The kernel sizes and paddings are configured to maintain constant spatial resolution throughout all layers, and for similar reasons, the channel size of all convolutional layer output feature maps is fixed at a constant value $C'$.

Except for the first layer, the LIFE module uses depth-wise separable convolutions (Chollet, 2017) for all other layers, which consist of a depth-wise convolution followed by a point-wise convolution. This type of convolution is efficient in terms of memory and computation, allowing the LIFE module to have minimal overhead compared to the original PFF layer. In addition, point-wise convolution can be used to process any number of auxiliary tokens, such as the classification token and the distillation token. The LIFE module is described in more detail in the following.

**Processing Patch Tokens in the LIFE Module.** The patch tokens $X_{patch} \in \mathbb{R}^{C \times N}$ are rearranged into an image format of shape $X_{p'} \in \mathbb{R}^{C \times H \times W}$, where $N = H \times W$. The $X_{p'}$ is then passed through a series of convolutional layers with progressively increasing receptive fields. Here, we use three layers:

$$
\begin{aligned}
F_{p_1} &= Conv_1(X_{p'}); \\
F_{p_2} &= Conv_2(F_{p_1}); \\
F_{p_3} &= Conv_3(F_{p_2});
\end{aligned}
\tag{3}
$$

where $F_{p_1}, F_{p_2}, F_{p_3} \in \mathbb{R}^{C \times H \times W}$ represent three different scales of local features obtained from the patch tokens. Each of these features is divided into three in the channel dimension and rearranged to form embeddings $Q_p, K_p, V_p \in \mathbb{R}^{C \times H \times W}$;

$$Q_p = \begin{bmatrix} F_{p_1}^{(C/3 \times H \times W)} \\ F_{p_2}^{(C/3 \times H \times W)} \\ F_{p_2}^{(C/3 \times H \times W)} \end{bmatrix}, \qquad K_p = \begin{bmatrix} F_{p_1}^{(C/3 \times H \times W)} \\ F_{p_2}^{(C/3 \times H \times W)} \\ F_{p_2}^{(C/3 \times H \times W)} \end{bmatrix}, \qquad V_p = \begin{bmatrix} F_{p_1}^{(C/3 \times H \times W)} \\ F_{p_2}^{(C/3 \times H \times W)} \\ F_{p_2}^{(C/3 \times H \times W)} \end{bmatrix}. \tag{4}$$

**Processing Auxiliary Tokens in the Proposed Module.** The class token $X_{cls} \in \mathbb{R}^{C \times 1}$ and any other auxiliary tokens can be processed using point-wise convolutions in the LIFE module:

$$\begin{aligned} F_{c_1} &= Conv\_1(X_{cls}); \\ F_{c_2} &= PointConv\_2(F_{c_1}); \\ F_{c_3} &= PointConv\_3(F_{c_2}); \end{aligned} \tag{5}$$

where $F_{c_1}, F_{c_2}, F_{c_3} \in \mathbb{R}^{C \times 1}$ represent different linear projections of the classification token for different scales. Next, each of the features is divided into three parts in the channel dimension and rearranged to form embeddings $Q_c, K_c, V_c \in \mathbb{R}^{C \times 1}$:

$$Q_c = \begin{bmatrix} F_{c_1}^{(C/3 \times 1)} \\ F_{c_2}^{(C/3 \times 1)} \\ F_{c_2}^{(C/3 \times 1)} \end{bmatrix}, \qquad K_c = \begin{bmatrix} F_{c_1}^{(C/3 \times 1)} \\ F_{c_2}^{(C/3 \times 1)} \\ F_{c_2}^{(C/3 \times 1)} \end{bmatrix}, \qquad V_c = \begin{bmatrix} F_{c_1}^{(C/3 \times 1)} \\ F_{c_2}^{(C/3 \times 1)} \\ F_{c_2}^{(C/3 \times 1)} \end{bmatrix}. \tag{6}$$

Subsequently, the features of the patch tokens $\in \mathbb{R}^{C \times H \times W}$ are flattened in spatial dimensions. The embeddings $Q$, $K$, and $V$ obtained from the patch tokens and the auxiliary tokens are concatenated to form the final embeddings $Q, K, V \in \mathbb{R}^{C \times (N+1)}$, which contain local features from different scales. The global information obtained through self-attention is complemented by the local context information encoded in these embeddings, resulting in improved performance. By combining global and local information, the model can better understand the context and relationships between different parts of the input.

## 4 Experiments

We analyze the LIFE module by integrating it into different state-of-the-art transformer architectures. The main experiments are conducted on small-scale image datasets, as our study focuses primarily on the performance of the transformer with limited data. We evaluate our model for the image classification task. However, many real-world applications are based on object detection or semantic segmentation, so we also examine how the module affects downstream dense prediction tasks. We demonstrate the effectiveness of our module for small datasets with quantitative and qualitative results in classification, detection, and segmentation tasks.

### 4.1 Experimantal Settings

**Datasets.** Small-scale datasets used in our experiments include CIFAR-100 (Krizhevsky et al., 2009), CIFAR-10 (Krizhevsky et al., 2009), TinyImageNet (Le & Yang), and ImageNet-100 (Deng et al., 2009). These datasets contain 50k, 60k, 100k, and 130k training samples, respectively. ImageNet-100 and TinyImageNet are subsets of the ImageNet-1k dataset (Deng et al., 2009), with 200 and 100 classes, respectively. CIFAR-10 and CIFAR-100 have 10 and 100 classes, respectively. An additional experiment is conducted on the standard large-scale dataset, ImageNet-1k, which contains 1,281,167 training samples. Except for the ImageNet-1k and ImageNet-100 datasets, all other datasets have small image resolutions, either $32 \times 32$ or $64 \times 64$. The VOC dataset (Everingham et al., 2010) is used for object detection, and the Cityscapes dataset (Cordts et al., 2016) is used for semantic segmentation tasks. The VOC and Cityscapes datasets contain 1,464 and 5000 samples for training, respectively.

**Implementation Details.** In our experiments, we used the Tiny and Small variants of the DeiT (Touvron et al., 2021), T2T-ViT-12, and Swin transformers as baseline models for the classification task. We obtained the LIFE variants of these models by replacing the linear projection that generates the query, key, and value with the LIFE module. We employ three scales with kernel sizes of 1, 3, and 5 and zero paddings of sizes of 0, 1, and 2, respectively.

We use the original image size in our experiments. For the Swin transformer, we select a window size of 8, and for T2T-ViT-12, we decrease the token dimension from 255 to 252 in order to process the token in a multiscale manner in the LIFE module for all datasets. The baseline models were designed for an input image size of 224×224. Therefore, we keep the network configurations the same as the baselines for the ImageNet-100 and ImageNet-1k datasets. Only for the Swin architecture, we resize the input to 256×256. For other datasets with small image sizes, we used a patch size of 1 for Swin and 4 for DeiT architectures. To adapt T2T-ViT-12, we replaced the first unfold operation with a 3×3 kernel with a stride of 2 and the last unfold operation with a 1×1 kernel.

For detection and segmentation tasks, we use a tiny version of DeiT and Swin as the backbone. In order to highlight the effect of the LIFE module, we employ simple heads for dense prediction tasks. For the object detection task, we exploit DETR (Carion et al., 2020) and evaluate different combinations of backbone and head with and without the LIFE module. For segmentation tasks, we use simple upsampling after the linear transformation operation as the segmentation head. The input image size for both tasks is 512×512.

For the classification, detection, and segmentation tasks, we followed the training details in (Touvron et al., 2021), (Anonymous, 2022), and (Zheng et al., 2021). We re-train all models from scratch with random initialization in our framework for a fair comparison. We use a batch size of 512 for small-scale datasets and 1024 for the ImageNet-1k dataset. For the dense prediction tasks, we also present results for ImageNet pre-trained initialization, as it is believed that the initialization can significantly impact the final performance.

Unlike transformer models, we observed that the ResNet (He et al., 2016) architecture benefited more from simple augmentations for small datasets. We, therefore, apply no augmentation other than random crop and random horizontal flip. ResNet models are trained with SGD with a momentum of 0.9 for 200 epochs. The initial learning rate is set to 0.1 and adjusted with a multi-stage scheduler, which multiplied the learning rate by 0.2 at epochs 60, 120, and 160.

## 4.2 Quantitative Results

In the following, we present the results of the evaluation of the LIFE module on various tasks and datasets. We first examine the efficacy of the LIFE module in addressing the performance gap between the transformer and the CNN counterpart when trained on smaller datasets in the image classification task. We then evaluate the versatility of the LIFE module by employing it for dense prediction tasks, including object detection and semantic segmentation.

### 4.2.1 Image Classification

We examine the efficacy of the LIFE module in addressing the performance gap between the transformer and the CNN counterpart when trained on smaller datasets. The LIFE module aims to address this issue by introducing locality bias into the transformer architecture through the use of convolutional layers. We train models on a small image classification dataset and compare their performance with that of CNNs, such as ResNet, of similar size. As shown in Table 1, the efficiency of the LIFE module is demonstrated by integrating it into multiple models with different sizes and architectures. We use the accuracy as a performance metric for comparison. The results show that the use of the LIFE module in the transformer architecture improves performance on small-scale datasets across different architectures and model sizes. The improvement is more significant for smaller model sizes; for instance, the LIFE module improves the performance of the DeiT-Tiny architecture by approximately 15% and the DeiT-Small architecture by approximately 5% on average of small dataset. We also evaluate the effect of using the LIFE module on ViTs on a large-scale image recognition dataset, and show that adding the LIFE module to different architectures preserves performance on the ImageNet-1k dataset.

Table 1: Performance comparison of ViTs on small-scale image classification datasets with and without the addition of the LIFE module. The results include Top-1 accuracy, number of parameters, and GMAC for DeiT and T2T architectures with an input size of 224×224×3, and for the Swin architecture with an input size of 256×256×3.

| | #Params | GMAC | IM-1K | CF-10 | CF-100 | Tiny-IM | IM-100 |
|---|---|---|---|---|---|---|---|
| CNN | | | | | | | |
| Resnet-18 | 11.23 | 2.37 | 70.53 | 94.14 | 75.10 | 60.86 | 80.74 |
| Resnet-50 | 23.71 | 5.35 | 76.63 | 94.32 | 74.25 | 63.45 | 82.70 |
| Transformers | | | | | | | |
| DeiT-T | 5.54 | 1.26 | 72.43 | 85.65 | 51.45 | 54.89 | 63.56 |
| DeiT-T-LIFE | 5.62 | 1.27 | 72.17 | **89.28** | **71.74** | **60.51** | **68.32** |
| T2T-ViT-12 | 6.66 | 1.74 | 75.93 | 88.41 | 52.51 | 57.89 | 83.58 |
| T2T-ViT-12-LIFE | 6.59 | 1.71 | 75.17 | **89.96** | **54.51** | **60.52** | **83.96** |
| DeiT-S | 21.70 | 4.61 | 80.00 | 87.24 | 70.39 | 55.08 | 80.84 |
| DeiT-S-LIFE | 21.85 | 4.64 | 79.35 | **90.38** | **73.97** | **59.78** | **81.52** |
| Swinv2-T | 27.72 | 5.95 | 81.39 | 95.03 | 76.65 | 64.78 | 86.86 |
| Swinv2-T-LIFE | 27.88 | 6.01 | 81.71 | **95.14** | **77.48** | **65.84** | **86.98** |

Although the T2T-ViT-12 and Swin transformers already incorporate some degree of locality through their modified architectural designs (i.e., the token-to-token module in T2T-ViT-12 processes overlapping windows; and the Swin has a pyramid architecture that combines neighboring tokens at each stage), the integration of the LIFE module further improves performance by providing multi-scale locality in every block. Overall, the improvement gain is more significant for DeiT, which lacks local information in its architecture, compared to the T2T and Swin Transformers.

We also report the number of parameters and GMACs required to infer an image size of 224×224 for T2T and DeiT, and 256×256 for the Swin transformer. The LIFE module has a negligible impact on the number of parameters and computations. In fact, T2T-ViT-12-LIFE is even more efficient than the baseline, as it has an embedding size of 252 compared to 256 for T2T-ViT-12.

Overall, these results demonstrate that the addition of the LIFE module improves the performance of ViTs on various small datasets and architectures. This effect is more prominent when the model is smaller and the dataset is more complex (with a higher number of classes and fewer samples per class). Additionally, the LIFE module can be easily integrated into different architectures with minimal memory and computational overhead.

### 4.2.2 Object Detection

For the object detection task, we use the DETR architecture (Carion et al., 2020), which consists of a feature extractor as the backbone and a transformer encoder-decoder (EncDec). In the encoder, the features extracted by the backbone are flattened and used as query $Q$, key $K$, and value $V$ in the self-attention layer of the encoder. In the decoder, the output of the encoder is used as $Q$ and $K$. $V$ is defined as a learnable parameter that is later used to predict the final details of the object.

To evaluate the effect of the LIFE module, we integrate it into both the backbone and the encoder-decoder. The DETR encoder-decoder does not include a linear transformation to generate $Q$, $K$, and $V$. To ensure a fair comparison, we first add a linear layer to the encoder-decoder just before self-attention to generate $Q$, $K$, and $V$ from the backbone features, which is denoted as EncDec-Linear. Then, we replace this linear layer with the LIFE module, referred to as EncDec-LIFE. We use the encoder-decoder without any transformation as a baseline, which is referred to as EncDec. We use the DeiT-T and Swinv2-T architectures and their LIFE variants as the backbone.

Table 2 shows the mAP and F1 scores for different combinations of the backbone and encoder-decoder. We observe that adding linear layers to generate $Q$, $K$, and $V$ leads to improvement. Replacing the linear layer with the LIFE module leads to additional improvement, indicating that the local information from

Table 2: Evaluation of the effectiveness of adding the LIFE module to ViTs in the object detection task using the DETR architecture with DeiT-T as the backbone, trained and tested on the VOC dataset.

| Backbone | Head | # params | GMAC | mAP | F1-Score |
|---|---|---|---|---|---|
| | Random Initialization | | | | |
| | EncDec | 27.20 | 14.20 | 27.31 | 37.52 |
| DeiT-T | EncDec-Linear | 28.38 | 14.50 | 29.76 | 40.24 |
| | EncDec-LIFE | 28.19 | 14.45 | **31.37** | **41.80** |
| | EncDec | 27.27 | 14.28 | 27.09 | 37.49 |
| DeiT-T-LIFE | EncDec-Linear | 28.46 | 14.58 | 29.22 | 39.60 |
| | EncDec-LIFE | 28.26 | 14.53 | **32.21** | **42.52** |
| | ImageNet-1k Initialization | | | | |
| | EncDec | 27.20 | 14.20 | 72.12 | 79.32 |
| DeiT-T | EncDec-Linear | 28.38 | 14.50 | 73.02 | 80.12 |
| | EncDec-LIFE | 28.19 | 14.45 | **73.51** | **80.42** |
| | EncDec | 27.27 | 14.28 | 72.43 | 79.68 |
| DeiT-T-LIFE | EncDec-Linear | 28.46 | 14.58 | 73.05 | 80.16 |
| | EncDec-LIFE | 28.26 | 14.53 | **73.92** | **80.73** |
| | EncDec | 50.07 | 27.32 | 78.21 | 84.48 |
| Swinv2-T | EncDec-Linear | 51.26 | 27.62 | 78.94 | 85.01 |
| | EncDec-LIFE | 51.06 | 27.57 | **80.57** | **86.13** |
| | EncDec | 50.23 | 27.57 | 80.24 | 85.92 |
| Swinv2-T-LIFE | EncDec-Linear | 51.41 | 27.88 | **81.27** | **86.73** |
| | EncDec-LIFE | 51.22 | 27.83 | **81.27** | 86.68 |

neighboring patches can aid in more accurate object detection in a scene. The best results are obtained when the LIFE module is used in both the backbone and the encoder-decoder.

We also evaluate the performance of the LIFE module with two different initializations. When the models are randomly initialized, the LIFE module improves performance by ∼+4 mAP. When ImageNet-1k pre-trained weights are used for initialization, the LIFE module improves performance by ∼+2 mAP. These results suggest that the inclusion of locality bias can be beneficial when training a model from scratch with a small dataset.

### 4.2.3 Semantic Segmentation

For the semantic segmentation task, we use a simple segmentation architecture to assess the effectiveness of the LIFE module. We use the DeiT-T and Swinv2-T architectures as the backbone and a simple linear layer followed by upscaling to match the dimension of the features to the number of classes as the segmentation head. DeiT-T-LIFE and Swinv2-T-LIFE refer to models in which the LIFE module is integrated into all the attention layers in the backbone.

Table 3 shows the performance in terms of the number of parameters, GMAC, and mIoU results for both models and two initialization methods. The LIFE module improves performance in both cases. When the model is initialized with random weights, the LIFE module improves performance by ∼2%. If ImageNet-1k pre-trained weights are used for initialization, the improvement is 0.55%. Similar to the object detection task, we observe a greater improvement when the model is trained from scratch.

### 4.3 Qualitative Results

To understand the decision-making process of a transformer model, we present the results of qualitative analyses using attention maps derived from transformer architectures. First, we generate attention visualizations for the classification task employing the existing *Attention Roll-Out* method (Abnar & Zuidema, 2020). Then, we propose the *Dense Roll-Out* method, which generates class-specific attention maps for dense

Table 3: Evaluation of the effectiveness of adding the LIFE module to ViTs on the semantic segmentation task using the DeiT-T architecture as the backbone, trained and tested on the Cityscapes dataset. The results include the mean IoU (mIoU), the number of parameters, and the computation cost (in GMAC).

| Model | # params | GMAC | mIoU |
|---|---|---|---|
| Random Initialization | | | |
| DeiT-T | 10.13 | 10.50 | 31.89 |
| DeiT-T-LIFE | 10.21 | 10.58 | **33.73** |
| ImageNet-1k Initialization | | | |
| DeiT-T | 10.13 | 10.50 | 58.62 |
| DeiT-T-LIFE | 10.21 | 10.58 | **59.17** |
| Swinv2-T | 32.36 | 23.95 | 59.37 |
| Swinv2-T-LIFE | 32.52 | 24.20 | **60.25** |

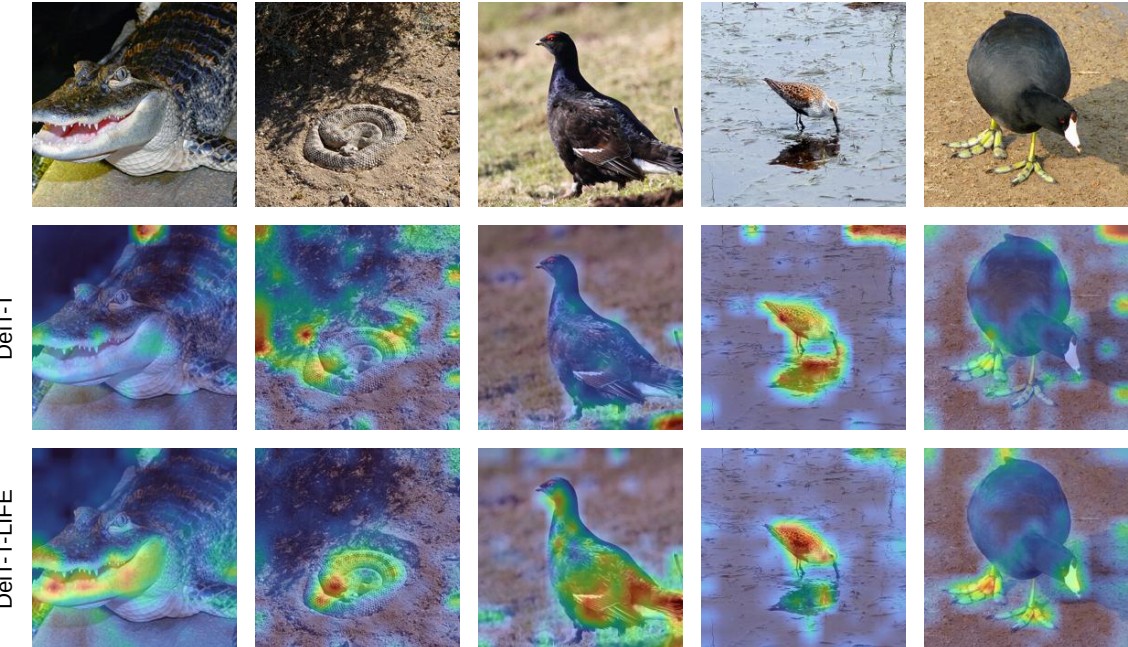

Figure 4: Comparison of attention maps of DeiT-T with and without the LIFE module, trained on the ImageNet-100 dataset, using the Attention Roll-Out method (Abnar & Zuidema, 2020).

prediction tasks, and demonstrate the effectiveness of the LIFE module utilizing our proposed visualization method.

### 4.3.1 Image Classification

We present a qualitative analysis of attention maps generated from DeiT-S with and without the LIFE module, trained on the ImageNet-100 dataset. As shown in Figure 4, the attention of a standard DeiT-T is scattered between the background and the foreground. However, when the LIFE module is used, the ViT architecture gains more local information, enabling it to better identify the foreground object and focus more on it.

### 4.3.2 Dense Prediction Tasks

**Dense Attention Roll-Out.** Attention mechanisms in transformer architectures allow the model to consider contextual information about the relationships between input tokens within a transformer block. At-

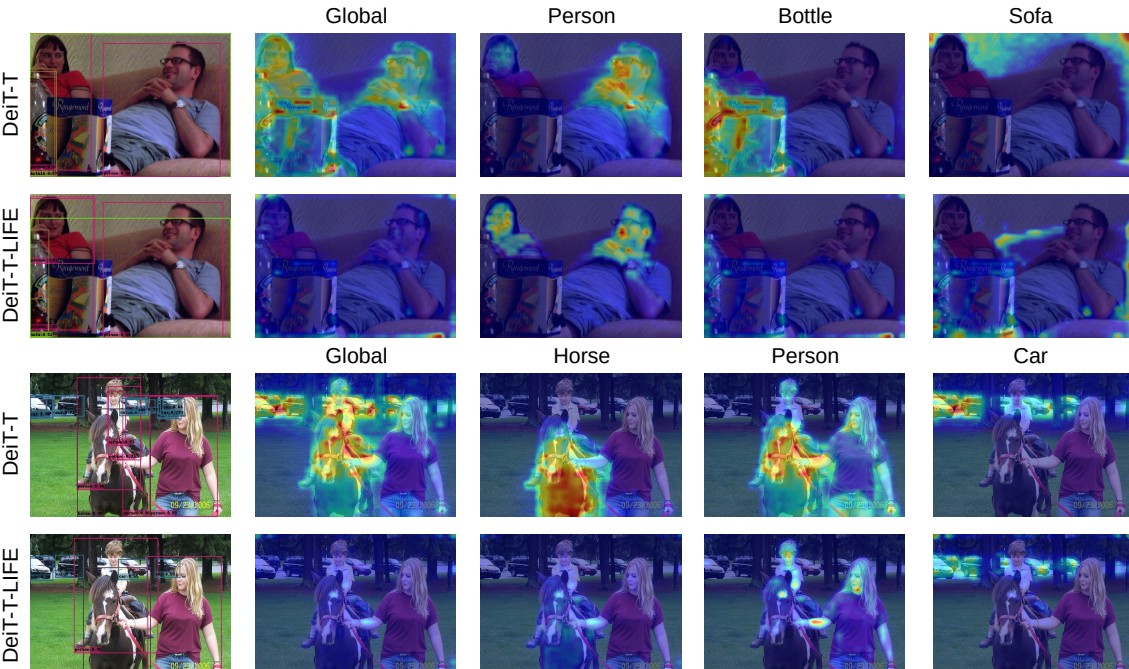

Figure 5: Comparison of attention maps of object detection models with and without the LIFE module using the proposed Dense Roll-Out method.

tention visualizations, which display the attention weights assigned to different input tokens, can provide insight into how the model makes its decisions and how well it can generalize to unseen data. There have been several methods proposed in the literature for generating attention visualizations in classification tasks, such as Attention Roll-Out (Abnar & Zuidema, 2020) and Gradient Attention Roll-Out (Chefer et al., 2021). These methods have been effective for classification tasks in vision transformers, but there is currently no method proposed for dense prediction tasks, such as object detection and semantic segmentation.

To address this gap, we propose a new method, *Dense Attention Roll-Out*, for generating attention visualizations in dense prediction tasks. This method is based on the Attention Roll-Out method, which creates a pairwise attention graph by linearly combining attentions from all blocks in a transformer architecture. However, we modified the method to specifically target dense prediction tasks. In contrast to the standard method, which only uses the attention map corresponding to the classification token, our method utilizes attention maps of all tokens, since all patch tokens are utilized for dense prediction tasks. Additionally, we use simple heads to generate attention maps using the features of the backbone, where most of the relevant information for prediction is located.

To generate class-specific attention maps, we use network predictions to identify relevant tokens for predicting a particular class by aligning the tokens spatially with the predictions, using either a segmentation map or a bounding box depending on the task. We then take the average of the attention maps for these corresponding tokens, remove the global content, and calculate the final class-specific attention map. The global content is common for all tokens that contain global information from the input image and is obtained by taking the average of all tokens.

**Result.** In order to demonstrate the effect of the LIFE module, we chose simple heads for both object detection and segmentation tasks. The use of simple heads allows us to generate visualizations using the backbone features, where the information for prediction is primarily located. Figures 5 and 6 show a comparison of models with and without the LIFE module in the backbone for object detection and segmentation tasks, respectively. The first three columns depict the input image, prediction, and global attention. The remaining columns show the class-specific attention maps, with the class names displayed at the top. The

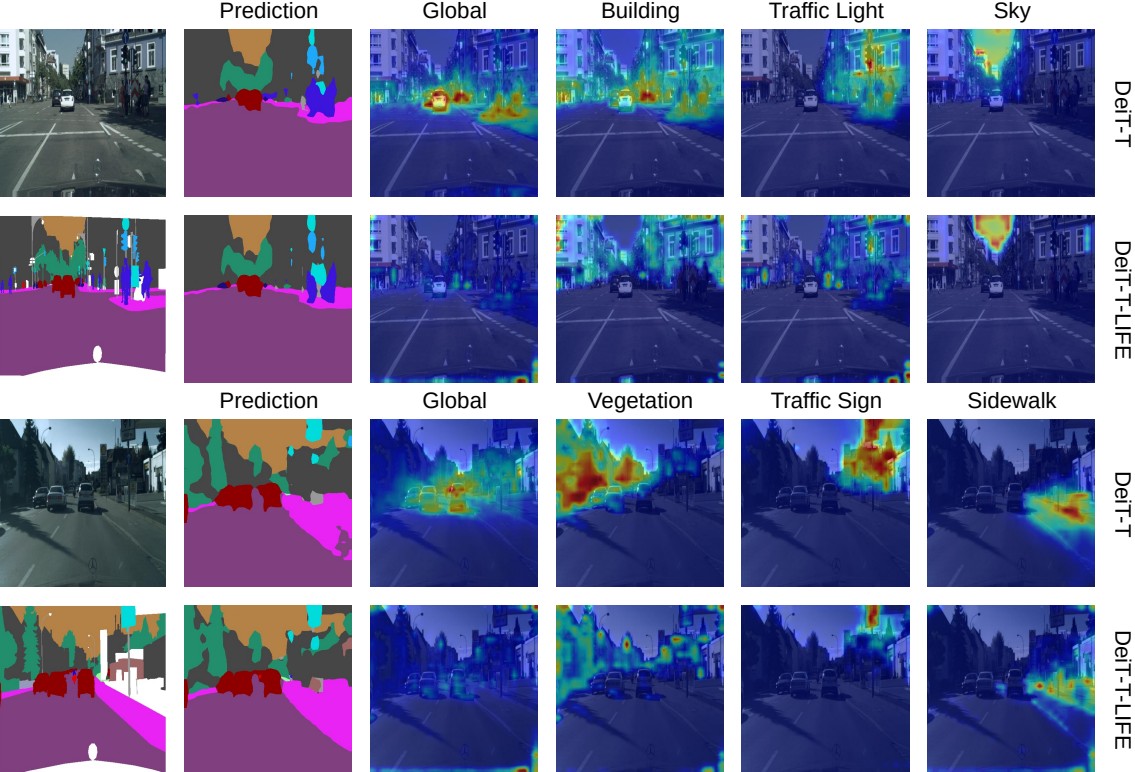

Figure 6: Comparison of attention maps of segmentation models with and without the LIFE module using the proposed Dense Roll-Out method.

attention maps illustrate that the LIFE module helps the model focus more on the relevant regions of the input image, leading to more accurate predictions.

## 5 Conclusion

We proposed a novel module, *Local InFormation Enhancer (LIFE)*, for vision transformers (ViTs) that effectively leverages local information from images to generate more informative embeddings for the self-attention layers in each transformer block. We evaluated the impact of the LIFE module on models with different sizes (DeiT-Tiny and DeiT-Small) and architectures (DeiT, T2T, and Swin). The classification results showed that while maintaining performance on large datasets, the proposed method consistently improved performance on small datasets. Additionally, we found that the improvement was more significant when the model had a lower capacity, indicating that the model effectively benefits from the locality bias introduced by the LIFE module. The LIFE module was also effective in the hierarchical transformer architecture, which inherently utilizes multi-scale information, suggesting that locality is important even at the window level. Furthermore, the incorporation of the LIFE module into ViTs for object detection and segmentation tasks improved performance, demonstrating the versatility and effectiveness of the LIFE module in various tasks. We also observed that initializing the model with parameters learned from the ImageNet dataset led to greater improvement compared to training the model from scratch, indicating that the local information encoded by the LIFE module helps ViTs achieve improved performance with minimal computational or memory overhead in a range of vision applications, including classification, object detection, and semantic segmentation. Finally, we introduced a new visualization method, Dense Attention Roll-Out, specifically tailored for dense prediction tasks such as object detection and semantic segmentation. This method allows for the generation of class-specific attention maps using attention maps of all tokens, providing insight into the decision-making process and generalization capabilities of a transformer architecture.

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

# A    Appendix

## A.1    Impact and Importance of Multiscale Embeddings

To demonstrate the effectiveness of multiscale information, we considered a modified version of the LIFE module called LIFE-OneScale, which encodes information using a single scale with a kernel size of three, similar to CvT. We evaluated this configuration by integrating it into DeiT-Tiny on the CIFAR-100 dataset. Using a single scale, we observed a 9.19% improvement over the baseline. However, encoding multiscale information with three kernels of sizes 1, 3, and 5 resulted in a 20.29% improvement over the baseline. These results emphasize the advantage of utilizing multiscale information.

Table 4: Comparison of performance on the CIFAR-100 dataset for models without locality (DeiT-T), with one-scale locality (DeiT-T-LIFE-OneScale), and with multi-scale locality (DeiT-T-LIFE).

| Model | Accuracy |
|---|---|
| Deit-T | 51.45 |
| Deit-T-LIFE-OneScale | 60.64 |
| Deit-T-LIFE | **71.74** |

## A.2    Comparison with Related Works

The design of several architectures, such as ConVit, Crossformer, and CvT, incorporates locality. This section aims to clarify the novelty of the LIFE model in comparison to these existing approaches.

ConViT utilizes a gated positional self-attention mechanism (GPSA) that integrates a positional self-attention component with a "soft" convolutional inductive bias. The self-attention block is enhanced by including positional information to achieve this objective. In contrast, LIFE alters the creation of query, key, and value embeddings, incorporating various convolutional operations to incorporate multiscale information into these embeddings before the self-attention operation.

Crossformer proposes two modules, namely the Cross-scale Embedding Layer (CEL) and the Long-Short Distance Attention (LSDA) modules, to incorporate locality into their design. The CEL module integrates various patches with different scales into each embedding to provide cross-scale features to the self-attention module. The LSDA module divides the self-attention module into short-distance and long-distance components to reduce computational load while maintaining small- and large-scale features in the embeddings. Our LIFE module shares a similar objective to the CEL module, as both leverage multiscale information. However, the LIFE module offers a more efficient and adaptable design compared to the CEL module. Specifically, the LIFE module utilizes depthwise separable convolution, which provides superior efficiency compared to the CEL module. Additionally, the LIFE module can handle auxiliary tokens, improving its flexibility for integration with diverse architectures. On the contrary, the Crossformer design does not incorporate auxiliary tokens, which prevents the CEL module from having this functionality.

The introduction of locality in CvT has been accomplished through two mechanisms: a convolutional patch embedding layer and a convolutional query, key, and value projection. The convolutional projection layer in CvT is similar to our own work; however, CvT employs only a single scale with a kernel size of three. In contrast, our LIFE module encodes multiscale information utilizing three scales with kernel sizes of 1, 3, and 5.

To perform a comparative analysis of the efficacy of the LIFE module with state-of-the-art approaches, we evaluated the performance of the CvT-21 model, which has 30 million parameters, on the CIFAR-100 dataset. Our results indicate that CvT-21 achieves an accuracy of 76.07%. The Swin Transformer with the LIFE module achieved an accuracy of 77.48%. This finding suggests that the LIFE module, in combination with a strong backbone, can outperform the CvT-21 model even with fewer parameters.

It is important to note that the objective of our study is to introduce locality into the transformer architecture in a modular and efficient manner. To achieve this goal, we proposed a LIFE module that efficiently leverages multi-scale information and adapts to a variety of architectures. We demonstrated its successful integration

into a diverse set of architectures, including the standard transformer architecture (Deit), the pyramid architecture (Swin), and a modified patch embedding of a standard transformer (T2T). Our findings indicate that the integration of the LIFE module effectively incorporates local information and leads to improved performance on small datasets.

### A.3   Limitations

### A.3.1   Performance on non-small datasets

Our method is highly effective for small-scale datasets, but we have found that it does not improve performance on larger datasets. The LIFE module is responsible for generating query, key, and value embeddings that contain an equal number of features across all scales. The resulting embeddings are concatenated to create the final query, key, and value tokens. Since the center of each scale is located at the same point, there may be some redundancy in the information encoded by different scales. Consequently, the representation capabilities of the query, key, and value tokens may be limited. Although the inductive bias provided by the LIFE module contributes to high performance in small-scale datasets, the limited embedding capacity adversely affects the module's ability to leverage locality, leading to similar performance in medium-scale datasets.

To test our hypothesis, we conducted a classification experiment on the ImageNet dataset, where we increased the size of the embeddings created by each scale in the DeiT-Tiny-LIFE architecture from 64 to 80. This resulted in an improvement in accuracy from 72.17 to 72.77, outperforming the baseline. However, this change also increased the number of parameters and the amount of GMAC, which conflicts with our objective of an efficient module. Furthermore, our study aims primarily to improve performance on small-scale datasets. For these reasons, we made a more efficient design choice, even though the performance remained constant in medium-scale datasets.

### A.3.2   Inference time

The LIFE module employs successive convolution operations to extract multiscale information. In order to apply the convolution operation to tokens, token reshaping is necessary. After processing, the resulting convolution outputs are concatenated along the channel dimension to generate a multiscale token. Although these reshaping, indexing, and concatenation operations require slightly more operations in terms of GMACS, they cause an increase in inference time because they are less efficiently implemented in PyTorch compared to linear projection. Consequently, models that incorporate the LIFE module exhibit slower performance than baselines.

We report the inference time of the models on both the GPU and the CPU in Table 5. In our experiments, we used an Intel(R) Core(TM) i7-8700 CPU operating at 3.20GHz and a GeForce RTX 2080 Ti GPU.

Table 5: Comparison of inference times for different models on GPU and CPU, using input sizes of 224×224×3 for DeiT and T2T architectures, and 256×256×3 for the Swin architecture.

| Model | GMACs | Inference Time (GPU) (ms) | Inference Time (CPU) (ms) |
|---|---|---|---|
| Resnet-18 | 11.23 | 2.86 | 16.42 |
| Resnet-50 | 23.71 | 6.91 | 38.30 |
| DeiT-T | 5.54 | 5.33 | 12.47 |
| DeiT-T-LIFE | 5.62 | 11.34 | 22.76 |
| T2T-ViT-12 | 6.66 | 6.59 | 29.34 |
| T2T-ViT-12-LIFE | 6.59 | 13.00 | 41.75 |
| DeiT-S | 21.70 | 5.13 | 34.73 |
| DeiT-S-LIFE | 21.85 | 12.22 | 49.73 |
| Swinv2-T | 27.72 | 11.04 | 77.14 |
| Swinv2-T-LIFE | 27.88 | 15.61 | 90.53 |

