# OpenReview forum: "LIFE: A Flexible and Efficient Module for Incorporating Local Information into Vision Transformers"
_TMLR — Rejected by TMLR_

### Review · Reviewer_cSoL · 2023-01-17

**Summary Of Contributions:**

This paper proposed a LIFE module to enhance the locality information for vision transformer, and the authors tested its capability in the vision tasks, including image classification, object detection and semantic segmentation, especially for smaller dataset.

**Audience:**

No

**Broader Impact Concerns:**

N/A.

**Claims And Evidence:**

No

**Requested Changes:**

See weaknesses. Especially the comparison to other methods that focuses on cooperating locality and how to further validate the proposed methods are good for the smaller dataset not the larger dataset.

**Strengths And Weaknesses:**

Strengths

1. The proposed LIFE module improves the standard vision transformer in all aspects.
2. The proposed module is light weights in terms of FLOPS and parameters.

Weaknesses

1. It is unclear to me how it is different from CvT and other similar works. The authors claim the differences from CvT is the LIFE module beneﬁts from multiscale information. However, there is no comparison in this regard and the LIFE module even degrades the performance on imagenet-1k

2. The proposed LIFE module degrades the performance of its counterpart for all different baselines in Table 1. There is no discussion or explanation on those. Moreover, there is no comparison to other methods that also try to cooperate locality information into ViT.
3. The authors claim the proposed module is better for small datasets, but I do not see any reason except for results in table 1.

4. For the results on object detection and semantic segmentation, there is no results built based on swin. As the results improved from swin is marginal in classification task, it will be better to show the result of swin in these two tasks.

5. A minor issue. Why call VT? I think the community call it ViT usually.

---

> ### Author Response · Authors · 2023-02-17
> **Response to Reviewer cSoL (1/3)**
>
> > It is unclear to me how it is different from CvT and other similar works.
>
> **CvT:** The introduction of locality in CvT is accomplished through two mechanisms: a convolutional patch embedding layer and convolutional query, key, and value projection. The convolutional projection layer in CvT is similar to our own work; however, CvT employs only a single scale with a kernel size of three. In contrast, our LIFE module encodes multiscale information by utilizing three scales with kernel sizes of 1, 3, and 5. To demonstrate the effectiveness of multiscale information, we implemented a modified version of the LIFE module, referred to as LIFE-OneScale, which encodes information using only a single scale with a kernel size of three, similar to CvT. We evaluated this configuration by integrating it into the DeiT-Tiny dataset on CIFAR-100 and our results indicate the importance of utilizing multiscale information.
>
> | Model | Accuracy |
> |--------------|:--------------:|
> | Deit-T | 51.45  |
> | Deit-T-LIFE-OneScale | 60.64 |
> | Deit-T-LIFE  | **71.74** |
>
> In addition to our experiments with DeiT-Tiny, we assessed the performance of the CvT-21 model, which has 30 million parameters, on the Cifar-100 dataset. Our results indicate that CvT-21 achieves an accuracy of 76.07%. The Swin Transformer with the LIFE module achieved an accuracy of 77.48%. This finding suggests that the LIFE module, in combination with a strong backbone, can outperform the CvT-21 model even **with fewer parameters**.
>
> **CrossFormer:** proposes the Cross-scale Embedding Layer (CEL) and the Long Short Distance Attention (LSDA) modules. CEL integrates numerous patches with various scales to each embedding to provide cross-scale traits to the self-attention module. In contrast, LSDA divides the self-attention module into two components, namely, short-distance and long-distance counterparts, to reduce the computational load while retaining both small-scale and large-scale features in the embeddings.
> The CEL module shares a similar objective with our LIFE module, as both leverage multiscale information. However, the LIFE module offers a more efficient and adaptable design compared to the CEL module. Specifically, the LIFE module utilizes depthwise separable convolution, which confers superior efficiency relative to the CEL module. Additionally, the LIFE module possesses a capacity to handle auxiliary tokens, thereby enhancing its flexibility for integration with diverse architectures. In contrast, the CrossFormer design does not incorporate auxiliary tokens, which precludes the CEL module from this functionality.
>
> **ConVit:** utilizes a gated positional self-attention mechanism (GPSA) that integrates a positional self-attention component with a "soft" convolutional inductive bias. The self-attention block is enhanced by the inclusion of positional information to achieve this objective. In contrast, LIFE alters the creation of the query, key, and value embeddings, incorporating various convolutional operations to incorporate multiscale information into these embeddings before the self-attention operation.
>
> We have added a new section to incorporate the comparison of our work with prior research (Section A.2).

---

> > ### Comment · Reviewer_cSoL · 2023-02-20
> > **response**
> >
> > > CvT: The introduction of locality in CvT is accomplished through two mechanisms...
> >
> > The comparison to CvT is very unclear to me. A more fair comparison is to replace the LIFE module with their convolutional query, key and value project on the same backbone. The comparison to CvT on Cifar-100 with Swin-T+LIFE only suggests that LIFE could further improve Swin-T, it does not indicate LIFE is better than CvT since the Swin-T itself has outperformed CvT-21 with fewer parameters (76.65 vs 76.07).
> > Moreover, when comparing the CvT-21 for ImageNet-1k (82.5), the Swin-T+LIFE is still worse (81.71).
> >
> > I think the comparison to those architectures also involve locality bias should be in the main manuscript as it is the main topic discussed in the paper.
> >
> > > To test our hypothesis, we conducted a classification experiment on the ImageNet dataset
> >
> > What is the parameters and GMACs increase after changing the embedding dimension from 64 to 80?
> >
> >
> > > One of the challenges of vision transformers is their high data requirements, which can be problematic when working with small scale datasets. To address this issue, researchers have proposed integrating a locality bias into the Vision Transformer architecture.
> >
> > Those architectures (CvT, ConViT, etc) also work well on the large datasets, I still do not get why the authors claims that the LIFE module is better in the small dataset.

---

> > > ### Author Response · Authors · 2023-02-27
> > > **Response to Reviewer cSoL (1/2)**
> > >
> > > > The comparison to CvT is very unclear to me. A more fair comparison is to replace the LIFE module with their convolutional query, key and value project on the same backbone.
> > >
> > > Thank you for your feedback. We apologize for any confusion caused by our previous comparison to CvT. Inline with your suggestion, we have replaced the LIFE module in the DeiT-T backbone with 3x3 depthwise separable convolution, as used in CvT. Our experiments on CIFAR-100 demonstrated that while the single-scale convolutional projection in CvT is competitive, our LIFE module outperforms it due to its ability to extract multi-scale features.
> > >
> > > | Model                      | Accuracy  |
> > > | -------------------------- | :-------: |
> > > | Deit-T                     |   51.45   |
> > > | Deit-T-ConvProjection(CvT) |   60.64   |
> > > | Deit-T-LIFE                | **71.74** |
> > >
> > >
> > > > The comparison to CvT on Cifar-100 with Swin-T+LIFE only suggests that LIFE could further improve Swin-T, it does not indicate LIFE is better than CvT since the Swin-T itself has outperformed CvT-21 with fewer parameters (76.65 vs 76.07).
> > >
> > >
> > > We agree that the comparison between Swin-T-LIFE and CvT was not intended to establish the superiority of our module over CvT. Instead, we aimed to show that our module can achieve state-of-the-art performance alongside a strong baseline. It is important to note that our LIFE module is designed to flexibly integrate into various architectures, not as a standalone architecture. Therefore, we cannot claim that LIFE outperforms CvT. However, our experiments do suggest that our module is more effective than CvT's convolutional projection layer in exploiting locality, as demonstrated on CIFAR-100.
> > >
> > > > Moreover, when comparing the CvT-21 for ImageNet-1k (82.5), the Swin-T+LIFE is still worse (81.71).
> > >
> > > Our main focus in this paper is to improve the performance of ViT on small-scale datasets, where traditional ViT architecture does not perform well due to the lack of inductive bias. We have demonstrated the effectiveness of our proposed LIFE module on small-scale datasets, where it outperforms the original ViT architecture. However, we understand the comparison made with CvT-21 and Swin-T+LIFE on ImageNet-1k. It's worth noting that the LIFE module produces multiscale embeddings, where each scale is centered at the same point. While this approach provides an advantage on small datasets with limited information, it may cause redundant information to appear across the various scales, which can limit the representation capabilities of query, key, and value tokens when training on larger and more complex data, such as the ImageNet-1k dataset. As a consequence, this leads to a decline in performance compared to other architectures designed for large-scale datasets.

---

> > > ### Author Response · Authors · 2023-02-27
> > > **Response to Reviewer cSoL (2/2)**
> > >
> > > > I think the comparison to those architectures also involve locality bias should be in the main manuscript as it is the main topic discussed in the paper.
> > >
> > > We appreciate the reviewer's suggestion to include comparisons to other architectures that involve locality bias in the main manuscript. However, as our proposed module is not a standalone architecture, we believe that such a comparison may not be appropriate or fair. Our focus is to demonstrate the effectiveness of our proposed module, and we performed experiments by integrating LIFE into various model architectures, presenting the results in our paper. This way, we aimed to show how our module can contribute to better results when dealing with limited amounts of data.
> > >
> > > Additionally, we have discussed related works that focus on locality bias in our paper. We also highlighted the limitations of our method, which include its inability to improve performance on large-scale datasets due to the presence of common information in the features extracted from different scales. We appreciate the reviewer's feedback and will ensure that our paper clearly communicates the contributions and limitations of our proposed method.
> > >
> > > > What is the parameters and GMACs increase after changing the embedding dimension from 64 to 80?
> > >
> > > When we increased the embedding dimension from 64 to 80, we observed an increase in the number of parameters from 5.62M to 6.36M, and the GMACs increased from 1.27 to 1.46. This is due to the larger token size resulting from the increased embedding dimension, which in turn leads to a more resource-intensive linear projection layer following the multi-head self-attention mechanism. We apologize for not providing this information in our original submission.
> > >
> > > > Those architectures (CvT, ConViT, etc) also work well on the large datasets, I still do not get why the authors claims that the LIFE module is better in the small dataset.
> > >
> > > We acknowledge that architectures, such as CvT and ConViT, work well on large datasets, and we did not claim that the LIFE module is superior to these methods on those datasets. Instead, our claim is that the LIFE module is better than convolutional projections on small datasets. This claim is supported by our experiments on smaller datasets including CIFAR-100 and CIFAR-10 datasets, which show that the LIFE module outperforms existing methods in terms of accuracy. While architectures such as CvT and ConViT are also effective on small datasets, they may not necessarily perform better than the LIFE module. The reason for this is that the LIFE module's inductive bias due to its multiscale nature, is well-suited for capturing locality, which is an essential feature of small datasets.

---

> ### Author Response · Authors · 2023-02-17
> **Response to Reviewer cSoL (2/3)**
>
> > The proposed LIFE module degrades the performance of its counterpart for all different baselines in Table 1.
>
> Our empirical findings demonstrate that the utilization of the LIFE module leads to enhanced performance across all 16 cases, which include four datasets and four models, for datasets of small size.
>
> An observed limitation of the LIFE module in enhancing performance is seen for the ImageNet dataset. The issue in performance can potentially be attributed to the following reason. The LIFE module is responsible for generating query, key, and value embeddings that contain an equal number of features across all scales. The resulting embeddings are concatenated to create the final query, key, and value tokens. Since each scale's center is located at the same point, there may be some redundancy in the information encoded by different scales. Consequently, the representation capabilities of the query, key, and value tokens may be limited. Although the inductive bias provided by the LIFE module contributes to high performance in small-scale datasets, the limited embedding capacity adversely affects the module's ability to leverage locality, leading to similar performance in medium-scale datasets.
>
> To test our hypothesis, we conducted a classification experiment on the ImageNet dataset, where we increased the size of the embeddings created by each scale in the DeiT-Tiny-LIFE architecture from 64 to 80. *This resulted in an accuracy improvement from 72.17 to 72.77, outperforming the baseline*. However, this change also increased the number of parameters and the amount of GMAC, which conflicts with our objective of an efficient module. Furthermore, the primary goal of our study is to improve performance in small-scale datasets. To this end, we made a more efficient design choice, even though it does not improve the performance in larger-scale datasets.
>
> To compare our module with alternative approaches that incorporate locality, we assessed the performance of the CvT-21 model, which has 30 million parameters, on the Cifar-100 dataset. Our results show that CvT-21 achieves an accuracy of 76.07%. The Swin Transformer with the LIFE module achieved an accuracy of 77.48%. This finding suggests that the LIFE module, in combination with a strong backbone, can outperform the CvT-21 model *even with fewer parameters*.
>
> On the other hand, the objective of our study is to formulate a module for incorporating locality into various transformer architectures. The chosen baseline models demonstrate the versatility of our module in being integrated into different architectures. These include the standard transformer architecture (Deit), pyramid architecture (Swin), and a modification to the patch embedding of a standard transformer (T2T). It should be noted that the selection of baseline models was not based on performance. Our module can be integrated to the state-of-the art method to introduce locality, potentially improving performance on smaller datasets.
> It is also important to note that we are not proposing a new network architecture, but rather an efficient module that can be integrated into existing architectures to improve performance when training with small datasets.

---

> ### Author Response · Authors · 2023-02-17
> **Response to Reviewer cSoL (3/3)**
>
> > The authors claim the proposed module is better for small datasets, but I do not see any reason except for results in table 1.
>
> One of the challenges of vision transformers is their high data requirements, which can be problematic when working with small scale datasets. To address this issue, researchers have proposed integrating a locality bias into the Vision Transformer architecture. This bias encourages the network to prioritize local features and relationships between nearby pixels, which improves performance on smaller datasets by reducing the need for large amounts of data.
> In light of this, the goal of this study is to incorporate the locality inductive bias into the transformer architecture. Results presented in Table 1 support the widely held belief that incorporating the locality inductive bias enhances data efficiency.
>
> Previous efforts to incorporate locality into vision transformer have employed a variety of approaches, but their solutions are either architecture-specific or computationally intensive. In this study, we focus on providing an efficient and modular solution. The improved performance on small-scale datasets provides evidence of the success of the proposed LIFE module in integrating the locality inductive bias.
>
> > For the results on object detection and semantic segmentation, there is no results built based on swin. As the results improved from swin is marginal in classification task, it will be better to show the result of swin in these two tasks.
>
> In accordance with review's suggestion, we evaluated LIFE module using the Swin architecture as a backbone for object detection and semantic segmentation tasks. Results indicated that, similar to the DeiT backbone, the addition of the LIFE module improved performance in comparison to the baseline backbone for both dense prediction tasks. Results are added to Tables 2 and 3.
>
> **Object Detection**
> | Backbone | Head | # params | GMAC | mAP | F1-Score |
> |--------------|:--------------:|:------------:|:------------:|:------------:|:------------:|
> | Swin_T | Standard | 50.07 | 27.32 | 78.21 | 84.48 |
> | | Linear | 51.26 | 27.62 | 78.94 | 85.01 |
> | | LIFE | 51.06  | 27.57 | 80.57 | 86.13 |
> | Swin_T_LIFE | Standard | 50.23 | 27.57 | 80.24 | 85.92 |
> | | Linear | 51.41 | 27.88 | 81.27 | 86.73 |
> | | LIFE | 51.22 | 27.83 | 81.27 | 86.68 |
>
> **Semantic Segmentation**
> | Model | # params | GMACs | mIoU |
> |--------------|:--------------:|:------------:|:------------:|
> | Swin-T | 32.36 | 23.95 | 59.37 |
> | Swin-T-LIFE | 32.52 | 24.20 | 60.25 |
>
> > A minor issue. Why call VT? I think the community call it ViT usually.
>
> The necessary revisions is implemented in the manuscript.

---

### Review · Reviewer_MZ68 · 2023-01-27

**Summary Of Contributions:**

This work presents LIFE - a convolution based local information incorporating module using depthwise separable convolutions that can be plugged into multiple Vision Transformer architectures to improve performance on smaller scale datasets. The authors show quantitative improvement on multiple smaller scale datasets and qualitative improvement on dense prediction tasks.

**Audience:**

Yes

**Claims And Evidence:**

Yes

**Requested Changes:**

Please compare with other works which aim to also make ViTs lightweight as mentioned above.

**Strengths And Weaknesses:**

**Strengths**

1. The results on small datasets are impressive with as little as 1% increase in computation
2. Interesting changes to the attention maps upon the addition of the local information

**Weaknesses**

There is a massive corpus of recently proposed lightweight vision transformer papers that the authors do not compare to which makes it difficult to gauge the results obtained. With other authors [Heo et al.] also proposing similar ideas, I would like to see a more thorough comparison with at least the following works:

- Park and Kim. How do vision transformers work?
- Wang et al. Pyramid vision transformers
- Heo et al. Rethinking Spatial Dimensions of Vision Transformers

---

> ### Author Response · Authors · 2023-02-17
> **Response to Reviewer MZ68**
>
> > comparing with more prior works
>
> The objective of our study is to introduce locality into the transformer architecture in a modular and efficient manner. To achieve this goal, we proposed a LIFE module and demonstrated its successful integration into various transformer-based architectures. Our experimental focus was on evaluating the contribution of the LIFE module to the performance of the models, rather than comparing it with existing architectures. It is important to note that we are not proposing a new network architecture, but rather an efficient module that can be integrated into existing architectures to improve performance when training with small datasets.
>
> We have added a new section to provide a more in-depth comparison of our work with prior research (see Section A.2).

---

### Review · Reviewer_PW5R · 2023-02-03

**Summary Of Contributions:**

This paper propose a efficient module for vision transformer, which can extract patch-level local information of input features. The proposed module can be inserted into various architectures to enhance performance. To verify the effectiveness of the method, the authors conduct experiments on various datasets, including  IM-1K and CF-10/100.

**Audience:**

Yes

**Claims And Evidence:**

Yes

**Requested Changes:**

See the weaknesses.

**Strengths And Weaknesses:**


Strenghts:

- The paper is well-written. The proposed method is simple and easy to follow.

- Inserting local information to a vision transformer is reasonable.

-Weaknesses:

- For Swinv2, the LIFE module improves performance marginally. For exmaple, Swinv2-Tiny achieves 95.03% acccuracy on CF-10, while  Swinv2-Tiny-LIFE achieves 95.14% (Table 1). It seems that LIFE produces limited improvement the hierarchical architectures. It is an interesting phonomenon. Could the authors analyze the reason?

- Only GMAC is reported in Table 1.  The running time is an important metric in practice. Will LIFE module bring latency cost significantly? The practical running time of different models is required in Table1.

---

> ### Author Response · Authors · 2023-02-17
> **Response to Reviewer PW5R**
>
> > For Swinv2, the LIFE module improves performance marginally.
>
> Hierarchical architectures possess a weak form of the locality inductive bias. This is because of the gradual combination of neighboring tokens, which allows the network to consider local context in its processing. As a result, the integration of the LIFE module to Swin have a comparatively lower impact in comparison to pure transformer architectures like DeiT.
>
> > Only GMAC is reported in Table 1. The running time is an important metric in practice. Will LIFE module bring latency cost significantly?
>
> The LIFE module employs successive convolution operations to extract multiscale information. In order to apply the convolution operation to tokens, token reshaping is necessary. Following processing, the resulting convolution outputs are concatenated along the channel dimension to generate a multi-scale token. Despite requiring slightly fewer operations in terms of GMACS, these reshaping, indexing, and concatenation operations are less efficiently implemented in PyTorch compared to linear projection. Consequently, models that incorporate the LIFE module exhibits slower performance than baselines when implemented in PyTorch. We incorporated these into "Limitations" Section.
>
> We employ an Intel(R) Core(TM) i7-8700 CPU operating at 3.20GHz and a GeForce RTX 2080 Ti GPU for performing inference. The results are reported in Section A.3.

---

### Review · Reviewer_h6C2 · 2023-02-05

**Summary Of Contributions:**

The paper proposes LIFE (Local InFormation Enhancer), an improved Q/K/V layer for ViTs. For each of Q/K/V, instead of computing these from token features using a linear layer (as is standard in Transformer), LIFE consists of three convolutions with differing spatial fields. This way, spatially local patches are included directly in the Q/K/V computation, which is an intuitively useful inductive bias for image models. Depthwise convs are used to keep computational overhead low. Experiments are performed on small classification datasets (CIFAR, etc.) using small models (up to DeiT-S size), plus ImageNet. Detection (VOC) and Semantic Segmentation (Cityscapes) are also included. LIFE provides a boost over DeIT/SwinV2/TVT, particularly in the smallest settings.

Overall, the paper makes three main claims:

(1) LIFE can be easily integrated w/ negligible memory/comp overhead.

(2) LIFE "consistently improves performance, regardless of the task at hand".

(3) Using (their new technique) dense attention rollout, qualitatively "LIFE can enhance local context learning by guiding the network to attend to more specific regions".


**Audience:**

Yes

**Broader Impact Concerns:**

None.

**Claims And Evidence:**

No

**Requested Changes:**

Please provide justification for the baselines used here, and why they appear quite weak compared to other results in the literature. Also, I think the claim about when LIFE helps e.g. "consistently improves performance, regardless of the task at hand"  should be reduced to cover the cases shown here (see weaknesses).

To improve interest in the paper, please consider comparison to similar approaches such as CvT and ConViT.

**Strengths And Weaknesses:**

Strengths

1. Claim 1 seems well substantiated, LIFE results in minimal overhead in terms of MACs and parameters. It also appears simple enough to implement easily.

2. The paper provides experiments on detection and semantic segmentation in addition to common classification tasks.

3. The paper adds LIFE to more than one architecture: vanilla ViT (DeIT), T2T-ViT, and Swin-v2.

4. On small-scale settings (dataset size, image resolution, architecture size), LIFE performs substantially better than the baseline without LIFE. 5-20% over DeiT-T on Cifar-10, 100, Tiny-ImageNet, and ImageNet-100.

Weaknesses

1. The baselines appear weak. The baselines appear to fall far behind models with similar cost in the literature, or much older models. While state-of-the-art is not a requirement for publication, this casts some doubt on the claim that LIFE consistently improves performance. For example:

On ImageNet, in [DeiT-iii](https://arxiv.org/abs/2204.07118), DeiT-S at 224x224 obtains 80.4% (from scratch) and [Swin-v2](https://arxiv.org/abs/2111.09883) reports 81.7% with Swinv2-T. These are 0.4% better than reported in here, while this difference is small, this difference is as great as the difference between LIFE & baseline (0.4% in the Swin-v2 case).

On CIFAR-10/100. CCT-7/3 ([Compact Convolutional Transformer](https://arxiv.org/abs/2104.05704v4)) reports 98% C10, 82.7% C100 w/ 3.76M params, 1.2 GMACs. In contrast, the best small baseline presented here obtains 88.4%/52.5%, C10/C100 respectively with 6.66M params, 1.74 GMACs: which is much larger and much lower performing. Even the larger models (~20 GMACs) perform much worse than CCT. The baseline in the CCT paper (ViT-Lite-7/4) obtains 93%/74%, C10/C100 respectively. Note for context: all of these numbers are very far off state-of-the-art for from-scratch training (e.g. ~90% on CIFAR100).

On detection/segmentation, baselines from 6-7 years ago perform substantially better than the baselines here: e.g. [YOLOv2](https://arxiv.org/abs/1612.08242v1) obtains 78.6 mAP on VOC detection, and [dilated ResNet](https://openaccess.thecvf.com/content_cvpr_2017/papers/Yu_Dilated_Residual_Networks_CVPR_2017_paper.pdf), 68 mean IoU with DRN-C-26. Here, the best numbers presented are 74 mAP VOC, 59 mIoU on Cityscapes. The fact that these models appear weaker than these old baselines may be due to some of the experimental design choices (e.g using a simple linear head in semantic segmentation). However, I believe this requires the claim that "boost in performance for dense prediction tasks" to be softened since it seems to only be substantiated in the case of a limited setup.

Overall, while the experiments do show benefits of LIFE, it would be useful to understand why the models seem weak in comparison to other models in the literature, or to soften the claims by restricting the scenarios where life provides a benefit.

2. LIFE appears to offer no benefit on ImageNet, therefore it seems only effective on the small scale (CIFAR/VOC/Cityscapes) data, which may limit interest.

3. Other papers include convolutions in the attention layer, such as [CvT](https://arxiv.org/abs/2103.15808) and [ConViT](https://arxiv.org/abs/2103.10697), it would be useful to see comparisons to these approaches.

4. The attention maps are interesting, however, with just a couple of hand-picked examples and no quantitative analysis, I think there is not sufficient evidence that "it is evident that the LIFE module helps the model focus more on the relevant regions of the input image"

Minor points

1. Some details are missing from the model description (e.g. position of layer norms).

2. VT -> ViT (to follow normal convention)

3. Nits

* typo: 244×224 -> 224x224 on p8
* "VTs do not, however, exploit this information due to a lack of local inductive bias" -> Soften this to "low local inductive bias" (there is still some in the first layer strided conv).

---

> ### Author Response · Authors · 2023-02-18
> **Response to Reviewer h6C2 (1/2)**
>
> > 1. The baselines appear weak.
>    >   - ImageNet
>
> In our experiments, we use the DeiT Pytorch implementation as the codebase and integrated the reported architectures into this framework. All methods were trained and evaluated within the same framework using a fixed seed. Any minor deviations from the official reported values may be due to the use of a different framework and/or seed selection.
>
> In the case of Deit-S, Touvron et al. propose a new training scheme in DeiT-iii, which includes the use of BCE loss, LAMB optimizer, and various augmentations. The number of epochs and batch size also differ from the first DeiT. As stated in Table 4 of their paper, with the use of cross-entropy loss, ViT-S achieves a score of 79.8, similarly our results indicate a score of 80.0 for the same metric.
>
>    >   - CIFAR-10/100
>
> The research conducted by Hassani et al. focuses on the design of a network architecture that exhibits high performance on small-scale datasets. They evaluate various hyperparameters for different architectures and find the optimal one for small-scale datasets. Their proposed pure transformer architecture (ViT-Lite-7/4) demonstrates accuracy of 93.57%/73.94% on CIFAR-10/CIFAR-100 without any modification to architecture.
>
> Our experiments, on the other hand, aims to demonstrate the modularity and efficacy of the LIFE module by integrating it into a diverse set of architectures, including the standard transformer architecture (DeiT), the pyramid architecture (Swin), and a modified patch embedding of a standard transformer (T2T). Our findings indicate that the integration of the LIFE module effectively incorporates local information and leads to improved performance on small datasets.
>
>   >   - detection/segmentation
>
> The weak performance of the baseline can be attributed to the selection of a simple head for object detection (DETR head) and semantic segmentation (simple linear projection). This decision was made to clearly demonstrate the contribution of the LIFE module, as it would not be evident if the head was utilizing the local information without this limitation.
>
> To supplement our existing experiments, we evaluated the performance of the LIFE module by incorporating it into the Swin architecture for both object detection and semantic segmentation tasks. The results indicate that the Swin backbone exhibits superior performance on these tasks (new results are added to Tables 2 and 3.). Table below shows that when LIFE module is integrated to both backbone and head, we observe a significant improvement regardless of the architecture.
>
> | Backbone | # params | GMAC | mAP | F1-Score |
> |------:|:------:|:------:|:------:|:------:|
> | DeiT-T | 27.20 | 14.20 | 72.12 | 79.32 |
> | DeiT-T-LIFE | 28.26 | 14.53 | **73.92** | **80.73** |
> | Swinv2-T | 50.07 | 27.32 | 78.21 | 84.48 |
> | Swinv2-T-LIFE | 51.22 | 27.83 | **81.27** | **86.68** |
>
> We have updated the paper to more accurately reflect the impact of the utilized settings on the observed improvement, in order to ensure the validity and integrity of our findings.
>
> > 2. LIFE appears to offer no benefit on ImageNet, therefore it seems only effective on the small scale (CIFAR/VOC/Cityscapes) data, which may limit interest.
>
> The LIFE module generates embeddings for queries, keys, and values that contain an equal number of features across all scales. The resulting embeddings are concatenated to create the final query, key, and value tokens. However, this may result in redundancy in the information encoded by different scales due to their centered position, limiting the representation capabilities of the tokens. Although the LIFE module's inductive bias performs well in small-scale datasets, the module's limited embedding capacity hinders its ability to leverage locality, leading to similar performance in larger scale datasets.
>
> To investigate this hypothesis, we conducted a classification experiment on the ImageNet dataset. We increased the size of the embeddings generated by each scale in the DeiT-T-LIFE architecture from 64 to 80 and observed an improvement in accuracy from 72.17 to 72.77, outperforming the baseline. However, this increase in the size of the embeddings also increased the number of parameters and the amount of GMAC, which conflicts with our objective of an efficient module. Furthermore, our primary objective is to enhance performance in small-scale datasets. For these reasons, we opted for a more efficient design choice, although the performance remains similar in larger scale datasets.

---

> ### Author Response · Authors · 2023-02-18
> **Response to Reviewer h6C2 (2/2)**
>
> > 3. Other papers include convolutions in the attention layer, such as CvT and ConViT, it would be useful to see comparisons to these approaches.
>
> **CvT:** The introduction of locality in CvT is accomplished through two mechanisms: a convolutional patch embedding layer and convolutional query, key, and value projection. The convolutional projection layer in CvT is similar to our own work; however, CvT employs only a single scale with a kernel size of three. In contrast, our LIFE module encodes multiscale information by utilizing three scales with kernel sizes of 1, 3, and 5. To demonstrate the effectiveness of multiscale information, we implemented a modified version of the LIFE module, referred to as LIFE-OneScale, which encodes information using only a single scale with a kernel size of three, similar to CvT. We evaluated this configuration by integrating it into the DeiT-Tiny dataset on CIFAR-100 and our results indicate the importance of utilizing multiscale information (Table below).
>
> | Model | Accuracy |
> |-----:|:-----:|
> | DeiT-T | 51.45 |
> | DeiT-T-LIFE-OneScale | 60.64 |
> | DeiT-T-LIFE | 71.74 |
>
> *In addition to our experiments with DeiT-Tiny, we assessed the performance of the CvT-21 model, which has 30 million parameters, on the CIFAR-100 dataset. Our results indicate that CvT-21 achieves an accuracy of 76.07%. The Swin Transformer with the LIFE module achieved an accuracy of 77.48%. This finding suggests that the LIFE module, in combination with a strong backbone, can outperform the CvT-21 model even with fewer parameters.*
>
> **CrossFormer:** proposes the Cross-scale Embedding Layer (CEL) and the Long Short Distance Attention (LSDA) modules. CEL integrates numerous patches with various scales to each embedding to provide cross-scale traits to the self-attention module. In contrast, LSDA divides the self-attention module into two components, namely, short-distance and long-distance counterparts, to reduce the computational load while retaining both small-scale and large-scale features in the embeddings.
> The CEL module shares a similar objective with our LIFE module, as both leverage multiscale information. However, the LIFE module offers a more efficient and adaptable design compared to the CEL module. Specifically, the LIFE module utilizes depthwise separable convolution, which confers superior efficiency relative to the CEL module. Additionally, the LIFE module possesses a capacity to handle auxiliary tokens, thereby enhancing its flexibility for integration with diverse architectures. In contrast, the CrossFormer design does not incorporate auxiliary tokens, which precludes the CEL module from this functionality.
>
> **ConViT:** utilizes a gated positional self-attention mechanism (GPSA) that integrates a positional self-attention component with a "soft" convolutional inductive bias. The self-attention block is enhanced by the inclusion of positional information to achieve this objective. In contrast, LIFE alters the creation of the query, key, and value embeddings, incorporating various convolutional operations to incorporate multiscale information into these embeddings before the self-attention operation.
>
> We have added a new section to incorporate the comparison of our work with prior works (Section A.2).
>
> > The attention maps are interesting, however, with just a couple of hand-picked examples and no quantitative analysis, I think there is not sufficient evidence that "it is evident that the LIFE module helps the model focus more on the relevant regions of the input image".
>
> We appreciate the reviewer's feedback and agree that our previous claim overstated the evidence presented. In our revised version, we have toned down the claim.
>
> > **Minor points**
>   >  1. Some details are missing from the model description (e.g. position of layer norms).
>
> Our experimental setup for the LIFE module entails a simple integration of the module in place of the linear projection that creates query, key, and value embeddings, with no further modifications to the baseline. Consequently, the network configuration remains consistent with the baseline.
>
> > 2. VT -> ViT (to follow normal convention)
> > 3. Nits
>
> We updated the paper accordingly.

---

### Author Response · Authors · 2023-02-17
**General Response**

We thank the reviewers for their valuable feedback and suggestions. We have diligently implemented the suggestions in the revised manuscript and conducted supplementary experiments. Here, we would like to address the common points:

One of the challenges of vision transformers is their high data requirements, which can be problematic when working with small scale datasets. It has been observed that convolutional neural networks (CNNs) outperform transformer architectures when trained with small datasets. This is attributed to the fact that the convolution operation introduces locality. This bias encourages the network to prioritize local features and relationships between nearby pixels, which improves performance on smaller datasets by reducing the need for large amounts of data. With this observation in mind, researchers have attempted to integrate locality into the transformer architecture. Although some methods have been proposed in the literature, they are either specific to certain network architecture or computationally inefficient.

Therefore, the objective of our study is to introduce locality into the transformer architecture in a modular and efficient manner. Our empirical results demonstrated the modularity and efficacy of our proposed LIFE module by integrating it into a diverse set of architectures, including the standard transformer architecture (Deit), the pyramid architecture (Swin), and a modified patch embedding of a standard transformer (T2T). Our module can be integrated to the state-of-the art method to introduce locality, potentially improving performance on smaller datasets. However, it is important to note that we are not proposing a new network architecture, but rather an efficient module that can be integrated into existing architectures to improve performance when training with small datasets.

The following modifications have been made to the revised manuscript:
- We expanded the experimentation on detection and segmentation tasks and incorporated additional experiments. We conducted a performance evaluation of the LIFE module by integrating it into the Swin architecture.  (**Tables 2 and 3**)

- To present an overview of the proposed method's limitations, we added **Section A.3** to the appendix. In this section, we specifically elucidated the constraints associated with inference time and ImageNet performance.

- We have added a new section to provide a more in-depth comparison of our work with prior research (**Section A.2**).

- We added an "impact and importance of multiscale embeddings" section to emphasize the benefits of utilizing multi-scale information. (**Section A.1**)

---

### Decision · Action_Editors · 2023-03-14

**Recommendation:** Reject

**Comment:**

The reviewers appreciate the generality and simplicity of the proposed approach and acknowledge its good performance on small scale datasets. However, after the authors' feedback, two reviewers remain unconvinced that the paper meets the acceptance threshold. The main reasons for this, stated in the reviewers' final recommendation, are:
- The lack of convincing comparisons to baselines. In particular, one reviewer was not convinced by the comparison with architectures incorporating a locality bias. The other reviewer felt, based on the authors' feedback, that using the hyper-parameters of the baselines out-of-the-box made the comparisons unfair, as it translates to comparing their well-tuned method to untuned ones.
- The lack of convincing explanations why the method is particularly well suited to small-scale datasets but not to large ones.
- The fact that the proposed method is a variant of existing components, which might limit the interest of the method to the community, in contrast to a broader study of ViT attention module variants.

Considering the above-mentioned points, and the lack of strong support from the other two reviewers, the AE believes that this paper is at best not ready for acceptance. Note that TMLR does not have "major revision" as a recommendation, hence the AE recommends rejection. However, if the authors strongly believe that they can convincingly address the reviewers' concerns, the AE would be willing to receive a revised version of the paper. This revised version would nonetheless go through another full round of reviews.

**Audience:**

The reviewers acknowledge the interest of the topic studied in this paper to the community, although one reviewer mentioned in their final review that the interest might be limited (see comments below).

**Claims And Evidence:**

The reviewers acknowledge that the claim of generality of the proposed method is convincingly demonstrated. However, several reviewers are not entirely convinced of the validity of the claim that the method yields a consistent improvement.